# Hybrid Self-Reinforced Composite Materials Based on Ultra-High Molecular Weight Polyethylene

**DOI:** 10.3390/ma13071739

**Published:** 2020-04-08

**Authors:** Dmitry Zherebtsov, Dilyus Chukov, Eugene Statnik, Valerii Torokhov

**Affiliations:** 1Center of Composite Materials, National University of Science and Technology “MISiS”, 119049 Moscow, Russia; dil_chukov@mail.ru (D.C.); vgtorohov@gmail.com (V.T.); 2Skolkovo Institute of Science and Technology, 143026 Moscow, Russia; Eugene.Statnik@skoltech.ru

**Keywords:** UHMWPE fibers, self-reinforced composites, single polymer composites, hybrid materials

## Abstract

The properties of hybrid self-reinforced composite (SRC) materials based on ultra-high molecular weight polyethylene (UHMWPE) were studied. The hybrid materials consist of two parts: an isotropic UHMWPE layer and unidirectional SRC based on UHMWPE fibers. Hot compaction as an approach to obtaining composites allowed melting only the surface of each UHMWPE fiber. Thus, after cooling, the molten UHMWPE formed an SRC matrix and bound an isotropic UHMWPE layer and the SRC. The single-lap shear test, flexural test, and differential scanning calorimetry (DSC) analysis were carried out to determine the influence of hot compaction parameters on the properties of the SRC and the adhesion between the layers. The shear strength increased with increasing hot compaction temperature while the preserved fibers’ volume decreased, which was proved by the DSC analysis and a reduction in the flexural modulus of the SRC. The increase in hot compaction pressure resulted in a decrease in shear strength caused by lower remelting of the fibers’ surface. It was shown that the hot compaction approach allows combining UHMWPE products with different molecular, supramolecular, and structural features. Moreover, the adhesion and mechanical properties of the composites can be varied by the parameters of hot compaction.

## 1. Introduction

The ultra-high molecular weight polyethylene (UHMWPE) is the main material used for the acetabular cup of artificial hip joints due to its outstanding physical and mechanical properties over other polymers. At the same time, different reports have a common conclusion that the UHMWPE is one of the leading contributors to failure in total joint replacements [1]. In other words, the generated debris due to wear particles and creep of the UHMWPE component results in joint loosening, osteolysis, and further required medical revision or even surgery [2,3]. Instead of total joint replacements, the UHMWPE is a widely used material for skis, neutron shielding, trabecular bone tissue replacement, and other industrial and consumer applications [4,5,6,7,8]. 

There are many approaches to the improvement in mechanical stability (hardness, elastic modulus), tribological behavior (wear resistance), or adhesive (roughness of surface) properties of the polyethylene. The most popular ones are the reinforcement with carbon nanotubes [9], graphene [10], kaolin or zeolite fibers [11,12]; filling of silver [13], zirconium/titanium/hafnium [14], alumina nanoparticles [15], or surface modification by argon plasma [16], chemical etching, electrostatic spraying techniques [17,18], and ultraviolet grafting [19]. On the other hand, many studies are focused on altering physical and chemical properties of the UHMWPE to increase wear resistance, using gamma-irradiated cross-linking [20,21], which correlates with decreases in wear. At the same time, this technique can result in a reduction of chemical stability, ultimate tensile strength, and impact strength of the UHMWPE [22]. Fiber reinforcement is also one of the promising methods for obtaining UHMWPE based composites with enhanced properties [23,24].

Many studies have proved that crystallinity plays an essential role in the tribological behavior of the UHMWPE. For instance, Karuppiah showed that an increase in crystallinity decreases the friction force and the scratch depth [25]. A similar result was reported by Wang, who demonstrated that the degree of crystallinity could be increased by controlling pressure and temperature during the material formation process [26]. Besides, there is a well-known fact that maximum crystallinity can be achieved only in fibers that have a highly oriented fibrillar structure. This structure can be achieved during gel-spinning followed by orientational drawing [27]. However, it is complicated to produce a bulk monolithic material from fibers due to the inertness of the UHMWPE. In 1975, Capiati and Porter presented a unique concept of reinforced polymeric materials. The idea was to produce a composite material where a reinforcement and a matrix should be made from the same polymer but with different morphologies [28]. Such composite materials are commonly called self-reinforced composite materials. As compared with traditional composites, in self-reinforced composite materials a strong and uniform fiber-matrix interface can be achieved because the problem of material difference is solved. Moreover, usually SRC has a low density compared to traditional composites based on carbon or glass fibers. This difference is achieved because polymeric fibers, as a reinforcement, have a lower density compared to traditional reinforcements (approximately 1.0 g/cm^3^ versus more than 2.0 g/cm^3^). In addition, the SRC can be easily recycled because there is no need to separate the composite components.

Different crystalline forms of amorphous-crystalline polymers (polymorphism), the existence of different supramolecular structures, or different grades of one polymer can be used to obtain the SRC. For example, a single-polymer composite can contain a tougher phase as a reinforcement and a less tough phase as a matrix. For example, β-polypropylene can be used as a matrix for α-polypropylene reinforcing elements [29]. However, there is a problem related to quite close melting points of the used materials, which can be solved using various types of the same polymer, for example, high- and low-density polyethylene (HDPE and LDPE), as a reinforcement and a matrix, respectively [30]. However, these materials do not align with the true concept of self-reinforcement and cannot be applied to biomedical applications [31]. In our previous paper, a method for the production of SRC materials based on UHMWPE fibers was proposed [32]. The developed technique is focused on hot compaction of UHMWPE fibers, which allows avoiding the relaxation processes resulting in a decrease in mechanical properties. Moreover, the proposed approach allows a more precise control of the melting process by shifting the melting point that occurs due to high hot-compaction temperatures [33]. 

Also, it is complicated to obtain a composite with complex shapes using hot compaction of UHMWPE fibers. Moreover, it is difficult to mold products using subtractive methods. Thus, it seems an economically and technically sound solution to make only a functional layer from the SRC. However, it is problematic to combine UHMWPE-based SRC with other materials due to the inertness of UHMWPE fibers. This problem can be solved using an isotropic UHMWPE as a part of the product. The isotropic part will not have a functional role as a self-reinforced composite material. This approach allows saving expensive UHMWPE fibers and makes it easier to process the product. An important property of such a composite is an adhesion at the interface between an isotropic UHMWPE and a self-reinforced composite material based on UHMWPE fibers. 

The mechanical, tribological, and other performance properties of the SRC were studied in our previous papers [34,35]. It was shown that in terms of the mechanical properties, the coefficient of friction, wear rate, and creep resistance, the designed SRC is far superior to the isotropic UHMWPE. The main goal of this study is to combine an isotropic and self-reinforced UHMWPE to create a multilayered single-polymer-based composite structure and investigate its mechanical and structural properties. 

## 2. Materials and Methods 

The UHMWPE powder grade GUR 4120 (molecular weight of approximately 5.0 × 10^6^ g/mol.) was used to produce an isotropic part of the multilayered sample. The powder was heated up to 180 °C at a pressure of 25 MPa in a stainless-steel mold to produce 80 × 10 × 2 mm^3^ rectangular samples. The UHMWPE fibers SGX (DSM Dyneema, Heerlen, The Netherlands) with an average diameter of 15 μm and a linear density of 220 Dtex were used for the preparation of the SRC. Hot compaction was used for producing rectangular samples with 80 × 10 × 3 mm^3^ in size for single-lap shear tests (SRC was 1 mm in thickness and isotropic UHMWPE was 2 mm in thickness). The fibers were wound on a special winding form to achieve the unidirectional state (Figure 1a). After that, the wound fibers were placed into a mold with isotropic UHMWPE plates and polytetrafluoroethylene (PTFE) plates. The PTFE plates allow separating the initial fibers and the isotropic UHMWPE in certain places as shown in Figure 1b. This separation allows controlling the contact area between the self-reinforced composite and the isotropic UHMWPE. The PTFE plates were removed after hot compaction and the samples for single-lap shear tests were obtained (Figure 1c). 

The size of the SRC and the isotropic UHMWPE contact (lap shear overlap) was approximately 10 × 10 mm^2^. A temperature ranges from 155 °C to 170 °C and a pressure of 25 MPa or 50 MPa were used for hot compaction. The heating time and the molding time were 50 min and 10 min, respectively.

The SRC’s rectangular samples of a size of 80 × 10 × 2 mm^3^ were obtained to determine the influence of the hot compaction parameters on the SRC’s properties using the same hot compaction parameters. The flexural test and the DSC analysis were carried out to determine the effect of hot compaction on the SRC’s structure and mechanical properties. 

Hot compaction allows melting a surface of each initial fiber (Figure 2a), the melted part after cooling forms a matrix of the SRC with an isotropic structure (Figure 2b). Moreover, the volume of the remelted UHMWPE fibers can be changed varying the temperature and the pressure during hot compaction.

The single-lap shear tests were carried out according to the standard test method for lap shear adhesion for fiber reinforced plastic bonding ASTM D5868-01 on a Zwick Z020 universal testing machine (Zwick Roell Group, Ulm, Germany) at a speed of 10 mm/min. The flexural tests were carried out on the same machine at a speed of 10 mm/min. For these tests, rectangular samples 80 × 10 × 2 mm in size were used. A NETZSCH DSC 204 F1 differential scanning calorimeter (NETZSCH Group, Selb, Germany) was used for a study of the SRC samples melting enthalpy. The measurements were carried out in argon atmosphere with the heating rate of 10 K/min according to ASTM D3417-83. A scanning electron microscope Hitachi TM-1000 (Hitachi, Marunouchi, Chiyoda-ku, Tokyo, Japan) was used to study the structure at a backscattered electron image mode. To conduct SEM observations, the samples were etched using the following mixture: one volume of orthophosphoric acid, two volumes of sulfuric acid, and 2% wt/vol of potassium permanganate. The samples’ surface was prepared by cutting in liquid nitrogen and etched for 4 h according to a method proposed in [36].

## 3. Results and Discussion

A highly oriented fibrillar structure of the initial UHMWPE fibers is a reason for their high mechanical properties in addition to a high degree of crystallinity (96.8% for the used fibers). The oriented structure is stable at an ambient temperature, while relaxation processes occur at temperatures close to the fibers’ melting temperature. These relaxation processes cause the disorientation of molecular chains, which results in the transformation from a fibrillar to lamellar structure with folded chain crystals. The lamellar structure of UHMWPE is mainly isotropic and shows poor mechanical properties as compared with the fibrillar structure. Moreover, this transformation causes a decrease in the crystallinity of UHMWPE. 

Crystallinity of the samples (*D_C_*) can be calculated as the relation of the area under the melting peak (sample’s melting enthalpy, Δ*H*) to enthalpy of melting of 100% crystalline sample, (Δ*H*^100^ = 291 *J*/*g*) [37]:
(1)$$D_{C}=\frac{\Delta H}{\Delta H^{100}}$$

Thus, the fraction of the preserved fibers and the isotropic UHMWPE (which was formed by melting of the fibers’ surface) can be calculated from the SRC melting enthalpy using the crystallinity of the fibers (*D_c_(fiber)*) and the isotropic UHMWPE (*D_c_(isotropic)*):(2)$$X(fiber)\times D_{C}(fiber)\times \Delta H^{100}+X(isotropic)\times D_{C}(isotropic)\times \Delta H^{100}=\Delta H,$$
where *X(fiber)* and *X(isotropic)* are fractions of the preserved fibers and the isotropic UHMWPE, respectively, and Δ*H*^100^ is the enthalpy of melting of 100% crystalline sample.

It was determined for the used UHMWPE fibers that crystallinity of the initial fibers and the remelted UHMWPE fibers was 96.8% and 54.4%, respectively. Thus Equation (2) can be rewritten for the used fibers:(3)X(fiber)×0.968×291[J/g]+X(isotropic)×0.544×291[J/g]=ΔH.

The DSC analysis was carried out for the SRC’s samples that were obtained at 25 MPa and at a temperature range from 155 °C to 170 °C with a step of 5 °C. The DSC analysis showed that the melting enthalpy was 267 *J*/*g*, 264 *J*/*g*, 252 *J*/*g*, and 200 *J*/*g* for the samples obtained at 155 °C, 160 °C, 165 °C, and 170 °C, respectively (Figure 3a). Using Equation (1), crystallinity was 92.0 ± 1.1%, 91.0 ± 1.3%, 87.0 ± 0.9%, and 69.0 ± 1.2% for the samples obtained at 25 MPa and at 155 °C, 160 °C, 165 °C, and 170 °C, respectively (Figure 3b). Thus, an increase in hot compaction temperature results in a decrease in SRC’s crystallinity caused by the fibers fraction reduction. Using Equation (3), the fraction of the preserved fibers was calculated as 88%, 85%, 76%, and 34% for the samples obtained at a temperature range from 155 °C to 170 °C, respectively. It is well known that for these materials, the DSC curves are characterized by two main peaks, the first of which (“high-temperature” peak) corresponds to the fibers and the other peak (“low-temperature” peak) corresponds to the matrix. It can be noted that the endoderm peaks at approximately 155 °C (“high-temperature” peak) decrease with an increase in hot compaction temperature. It also confirms that the preserved fibers’ content reduces with an increase in temperature of hot compaction [38]. Moreover, the small peak at a lower temperature can be observed for the samples obtained at 170 °C. It is related to the appearance of a high amount of matrix phase that was caused by intense fiber melting at high processing temperatures. 

The single-lap shear test was carried out to determine the influence of hot compaction temperature on shear strength between the SRC and the isotropic UHMWPE. An increase in hot compaction temperature results in an increase in shear strength, which were 2.10 ± 0.10 MPa, 2.20 ± 0.11 MPa, 2.90 ± 0.12 MPa, and 3.80 ± 0.14 MPa for the samples obtained at a pressure of 25 MPa and at 155 °C, 160 °C, 165 °C, and 170 °C, respectively (Figure 3b). 

An increase in shear strength is caused by better consolidation between the isotropic UHMWPE and the remelted fiber fraction that increases with an increase in hot-compaction temperature. However, crystallinity of the SRC decreases from 92.0 ± 1.1% to 69.0 ± 1.2% for the samples obtained at a temperature range from 155 °C to 170 °C, respectively. Consequently, the higher temperature causes an increase in shear strength while the fiber fraction decreases and mechanical properties of the SRC reduce, which was proved by the flexural tests. Moreover, the obtained shear strength is slightly higher as compared with the laminated sheets of UHMWPE obtained at 132 °C and 13.8 MPa by hot compaction with an polyurethane matrix (range of 2.1–3.8 MPa in this paper, against approximately 2 MPa in [39]).

The flexural strength and the flexural modulus were determined depending on hot compaction parameters for the SRC’s samples obtained at 25 MPa and different temperatures (Figure 4). It was shown that the flexural strength increases from 110.2 ± 1.5 MPa to 117.3 ± 1.7 MPa with increasing hot compaction temperature, which causes an increase in the matrix content. An increase in the matrix content results in a better load transfer to reinforcing elements. However, the flexural modulus decreases from 37.7 ± 1.8 GPa to 18.0 ± 1.4 GPa for the samples obtained at a temperature range from 155 °C to 170 °C, respectively. This behavior is caused by a decrease in the preserved fibers content and the relaxation processes that accompanied a decrease in the stiffness of the composites [40]. 

Also, the shear test was carried out for the samples obtained at 50 MPa and at a temperature range from 155 °C to 170 °C with 5 °C step. The photos of the samples are shown in Figure 5a,b. The shear strength for these samples has a similar increasing behavior as for the samples obtained at 25 MPa. The shear strength was 1.10 ± 0.12 MPa, 1.40 ± 0.16 MPa, 2.30 ± 0.18 MPa, and 2.90 ± 0.09 MPa for the samples obtained at 155 °C, 160 °C, 165 °C, and 170 °C, respectively (Figure 5c,d). It can be seen that the shear strength for the samples obtained at a higher pressure (50 MPa) is lower as compared with the samples obtained at a pressure of 25 MPa and at the same hot compaction temperature range. This difference is caused by lower fibers’ remelting in the samples obtained at a higher pressure and at the same hot compaction temperature. The lower fibers’ remelting was also proved by the DSC analysis, which showed an increase in crystallinity for the samples obtained at a pressure of 50 MPa as compared with crystallinity of the samples obtained at 25 MPa. Crystallinity decreases from 94.0 ± 0.9% to 84.0 ± 1.7% with an increase in hot compaction temperature for the samples obtained at 50 MPa (Figure 5d); whereas crystallinity for the samples obtained at 25 MPa decreases from 92.0 ± 1.1% to 69.0 ± 1.2 (Figure 3b). Table 1 presents all the values of the shear strength and crystallinity for the samples manufactured at a pressure of 25 or 50 MPa and temperatures from 155 to 170 °C.

To study the composites’ cross-section, the samples were cut in the direction perpendicular to the fibers’ axis. The cutting was made in liquid nitrogen and the followed etching was made to show the structural features of the composites. The surfaces were studied using scanning electron microscopy (Figure 6). Two top pictures show the structure of the SRC’ samples obtained at 155 °C and at a pressure of 25 MPa (Figure 6a) or 50 MPa (Figure 6b). The potassium permanganate etching is more effective for the isotropic UHMWPE. Thus, the isotropic UHMWPE matrix, formed as a result of the fibers’ surface melting, was etched more effectively than the preserved part of the fibers. Thus, the etching grooves characterize the thickness of the remelted fibers layer, and depending on hot compaction parameter, it changes in a range of 1–3 μm. It can be seen that due to the high processing pressure, the transverse section of the fibers becomes nearly ellipse. The melted UHMWPE fills the gaps between the fibers and forms a uniform, monolithic composite material. The amount of the melted UHMWPE is sufficient to connect the fibers with each other. The fibers in the SRC obtained at a lower pressure have a rounder shape than that of at higher pressure. The etching grooves are wider for the samples produced at a lower pressure, which means higher fibers’ remelting at pressure. It was also confirmed by the DSC analysis (Figure 7). Crystallinity was 87 ± 0.9% and 89 ± 1.3% for the samples obtained at 25 MPa and 50 MPa, respectively. 

Also, the interfaces between the self-reinforced composites based on UHMWPE and the isotropic UHMWPE are shown in Figure 6c,d. These hybrid composites were obtained at 165 °C and at a pressure of 25 MPa (Figure 6c) or at 50 MPa (Figure 6d). It can be seen that the fibers penetrate into the isotropic UHMWPE and form a uniform interface without gaps and structural defects. Such an interface provides an effective load transfer between the layers and results in relatively high interlayer shear strength. 

The DSC curves were obtained for the samples obtained at 165 °C and a pressure of 25 MPa or 50 MPa (Figure 7) to compare the dependence of crystallinity on the content of the preserved fibers. According to the Clausius–Clapeyron relation, the melting temperature increases with an increase in pressure [33]. Thus, higher pressure allows preserving higher volume of each UHMWPE fiber. For example, crystallinity was 87% and 89% for the samples obtained at 25 MPa and 50 MPa, respectively. 

Thus, the proposed approach to obtaining the SRC provides various ratios of a matrix to the oriented phase. This allows obtaining composites for various fields of application, the mechanical and other functional properties of which will be determined by the technological parameters of their production, which allows varying the properties of materials depending on the requirements for their application.

## 4. Conclusions

An approach to obtaining UHMWPE-based two-layered composite materials was proposed, and the structure and mechanical properties of the composites were studied. The first layer of the composites was an isotropic UHMWPE, and the other one was a self-reinforced unidirectional composite material based on UHMWPE fibers. Hot compaction allowed creating a hybrid material formed by the surface melting of the UHMWPE fibers and the remelted UHMWPE formed a matrix of the SRC and bound the SRC and the isotropic layer. 

The shear strength between two layers was 2.10 ± 0.10 MPa, 2.20 ± 0.11 MPa, 2.90 ± 0.12 MPa, and 3.80 ± 0.14 MPa for the samples obtained at a pressure of 25 MPa and at temperatures of 155 °C, 160 °C, 165 °C, and 170 °C, respectively. A similar trend was observed for the samples obtained at a pressure of 50 MPa and the same conditions. The shear strength increased from 1.10 ± 0.12 MPa to 2.90 ± 0.09 MPa for the samples obtained at a temperature range from 155 °C to 170 °C, respectively. This behavior was caused by an increase in the matrix content in the SRC, which bonds the isotropic and SRC layers. Crystallinity, as a direct indicator of the fiber content, was 92.0 ± 1.1%, 91.0 ± 1.3%, 87.0 ± 0.9%, and 69.0 ± 1.2% for the samples obtained at a pressure of 25 MPa and temperatures from 155 °C to 170 °C, respectively. Thus, the adhesion between the layers increased with an increase in matrix content, which is caused by an increase in hot compaction temperature. However, higher temperatures resulted in lower fiber content that was accompanied with poorer SRC’s properties. It was shown that the SRC’s flexural modulus decreases from 37.7 ± 1.8 GPa to 18.0 ± 1.4 GPa with increasing hot-compaction temperature. However, an increase in the matrix content allowed a slight increase in the flexural strength of the SRC (from 110.2 ± 1.5 MPa to 117.3 ± 1.7 MPa), which was related to a better load transfer to the preserved fibers. 

The SEM examination was used to study the structure of the hybrid and self-reinforced composites based on UHMWPE. Using a special etching approach to visualization, it was shown that the hot compaction allows melting only the surface of each fiber. The amount of the remelted UHMWPE fibers is sufficient to fill the gaps between the fibers. Also, it can be noted that in the hybrid composites, the fibers penetrate into the isotropic UHMWPE and form a large surface contact between the SRC and the isotropic UHMWPE layers. Such an interface provides an effective load transfer between the layers and results in relatively high interlayer shear strength.

## Figures and Tables

**Figure 1 materials-13-01739-f001:**
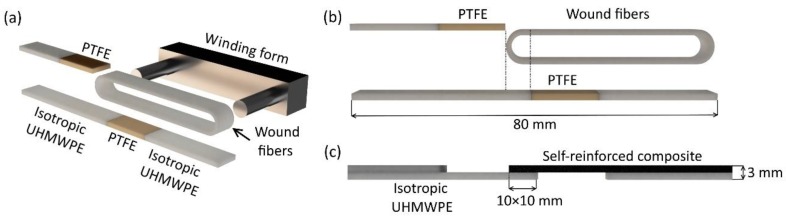
Scheme of obtaining the hybrid composite material based on ultra-high molecular weight polyethylene (UHMWPE) for the single-lap shear tests. The winding of initial UHMWPE fibers (**a**) and parts of hybrid composite material before (**b**) and after (**c**) hot compaction.

**Figure 2 materials-13-01739-f002:**
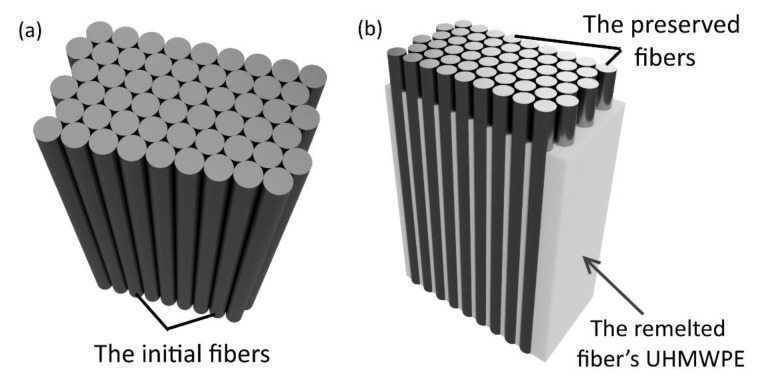
Scheme of the self-reinforced composite (SRC) preparation: initial UHMWPE fiber (**a**) and self-reinforced composites after hot compaction (**b**).

**Figure 3 materials-13-01739-f003:**
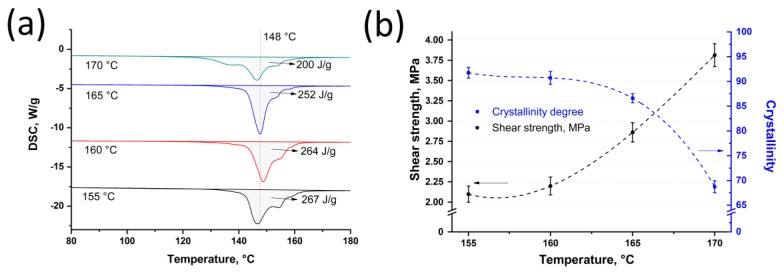
Results of differential scanning calorimetry (DSC) curves (**a**) and shear strength of hybrid composites and crystallinity (**b**) of the self-reinforced composites based on UHMWPE produced at 25 MPa and various temperatures.

**Figure 4 materials-13-01739-f004:**
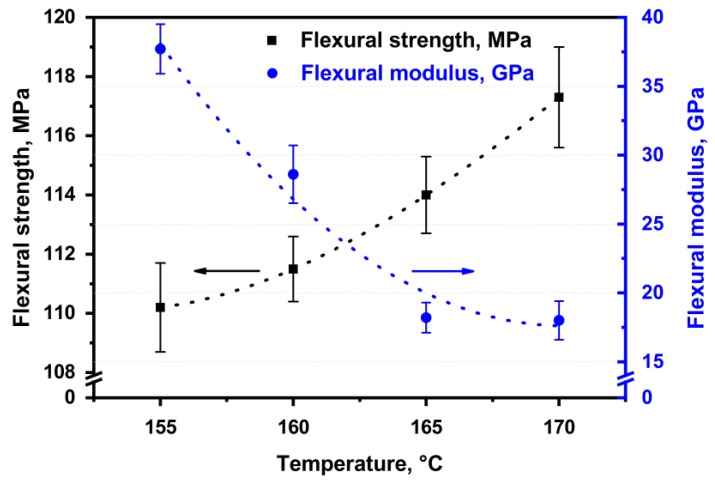
Flexural strength and flexural modulus of the self-reinforced composite based on UHMWPE produced at 25 MPa and different temperatures.

**Figure 5 materials-13-01739-f005:**
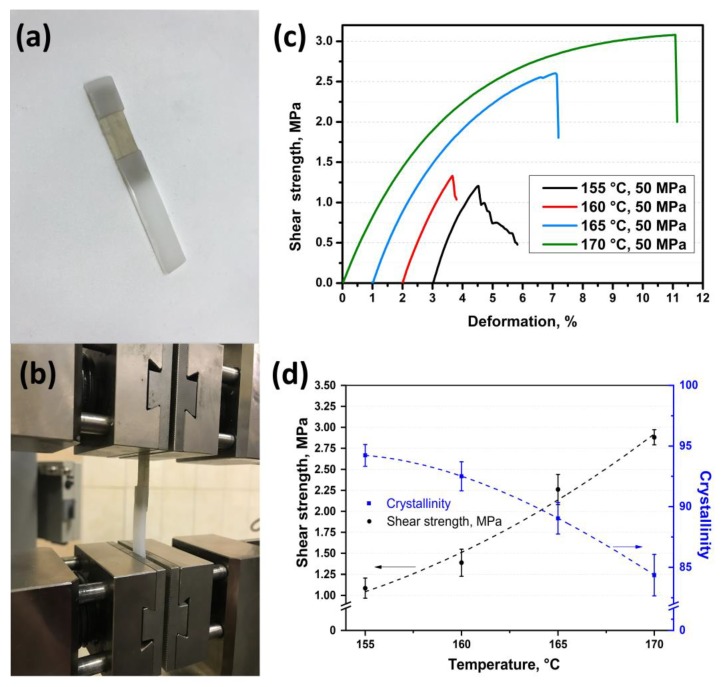
Photo of the samples (**a**,**b**), stress-strain curves (**c**) and shear strength (**d**) of the hybrid composites based on UHMWPE produced at 50 MPa and various temperatures.

**Figure 6 materials-13-01739-f006:**
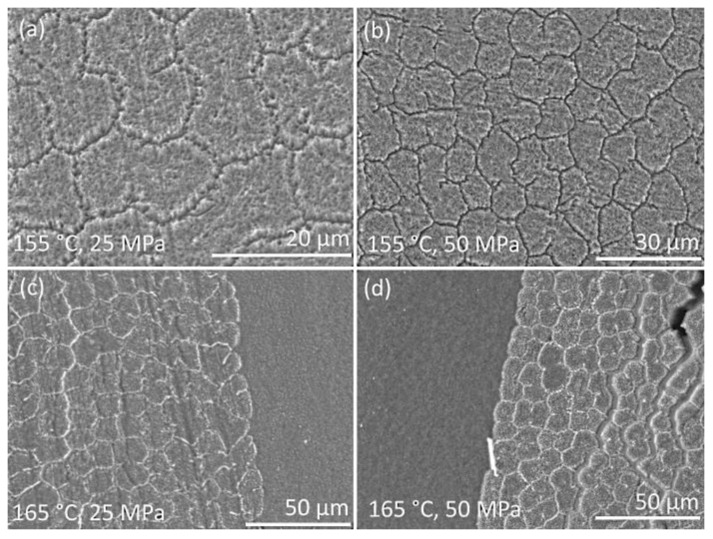
SEM images of the SRC’s and hybrid composites’ surface of samples obtained at 155 °C, 25 MPa (**a**), 155 °C, 50 MPa (**b**), 165 °C, 25 MPa (**c**) or 155 °C, 25 MPa (**d**).

**Figure 7 materials-13-01739-f007:**
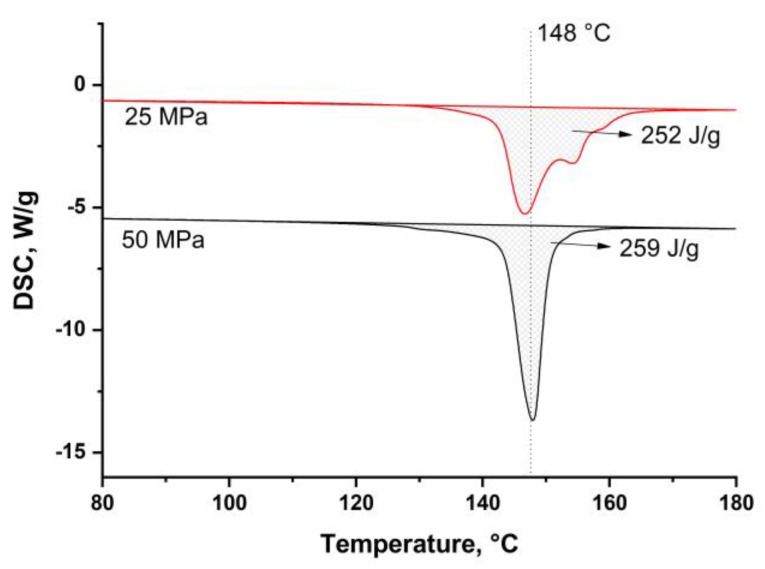
DSC analysis of the samples obtained at 165 °C and pressure of 25 MPa (red curve) and 50 MPa (black curve).

**Table 1 materials-13-01739-t001:** Shear strength and crystallinity of the SRC obtained at different temperatures and pressures.

Hot Compaction Temperature, °C	25 MPa		50 MPa	
	Shear Strength, MPa	Crystallinity, %	Shear Strength, MPa	Crystallinity, %
155	2.10 ± 0.10	92.0 ± 1.1	1.10 ± 0.12	94.0 ± 0.9
160	2.20 ± 0.11	91.0 ± 1.3	1.40 ± 0.16	93.0 ± 1.2
165	2.90 ± 0.12	87.0 ± 0.9	2.30 ± 0.18	89.0 ± 1.3
170	3.80 ± 0.14	69.0 ± 1.2	2.90 ± 0.09	84.0 ± 1.7

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
