# Peer review of "Hybrid Self-Reinforced Composite Materials Based on Ultra-High Molecular Weight Polyethylene"

_materials, 2020, doi:10.3390/ma13071739_

Round 1
Reviewer 1 Report
The paper by Zherebtsov describes interesting composite materials based on ultra-high molecular weight polyethylene. The results are clearly presented but characterization of the materials is insufficient. It should be extended by further common experimental techniques. For example, X-ray diffraction and infrared spectroscopic data should be very helpful for this type of composites as both provide important information about crystallinity of polyethylene.
Minor points:
- Units “g/J” should not be given in equations 2 and 3.
- Line 191: “50 MPa” or “25 MPa”?
- Data presented experiments at 25 and 50 MPa and different temperatures should be listed in one table. It should improve readability considerably.
Author Response
Response to Reviewer 1 Comments
The paper by Zherebtsov describes interesting composite materials based on ultra-high molecular weight polyethylene. The results are clearly presented but characterization of the materials is insufficient. It should be extended by further common experimental techniques. For example, X-ray diffraction and infrared spectroscopic data should be very helpful for this type of composites as both provide important information about crystallinity of polyethylene.
Minor points:
- Units “g/J” should not be given in equations 2 and 3.
- Line 191: “50 MPa” or “25 MPa”?
- Data presented experiments at 25 and 50 MPa and different temperatures should be listed in one table. It should improve readability considerably.
Response 1: Please provide your response for Point 1. (in red)
Dear reviewer, thank you for carefully reading of the paper and your helpful suggestions. We corrected the paper according to your comments; see below for a more detailed description.
We absolutely agree that the article can be more informative and complete if it contains the X-ray, IR or Raman spectroscopy studies. We have started to work with such techniques to study our samples. But it is not finished work as it take a lot of time. For now, we got the WAXS pattern for the initial fibers and our samples (figure 1).
Figure 1 – WAXS patterns from (a) initial fibers, samples, obtained at 25 MPa and at (b) 155 °C or (c) 165 °C
It can be seen that the pattern of initial fibers (fig 1(a)) is tight for each plane. It means that the crystals into the initial fibers have a high orientation. It can be noted that the patterns increase with an increase in temperature of hot compaction. Thus, the high temperature results in disorientation of crystals in SRC’s samples. This fact shows an increase in remelted fibers with an increase in temperature of hot compaction. These results correlate with our DSC analyses. There are some approaches which allows to calculate the crystallinity of studied samples from the WAXS image. The orientation of crystalline planes and molecules in crystals also can be calculated from WAXS. These technics were showed in [Shavit-Hadar, Liron, Dmitry M. Rein, Rafail Khalfin, Ann E. Terry, Guido Heunen and Yachin Cohen. “Compacted UHMWPE fiber composites: Morphology and X‐ray microdiffraction experiments.” (2007)].
Moreover, we started to work with Raman spectroscopy. However, this technic requires deep data processing. At the moment, we estimated that the crystallinity decreased with an increase in hot compaction temperature.
WAXS and Raman spectroscopy are quite informative and difficult to be calculated and take a lot of time. We suggest that these results should be issued in a new article. The current article is aimed to describe the conception and methodology of the obtaining of hybrid composite materials based on UHMWPE. The main goal of this article is to study mechanical properties and interface between components of the hybrid composites. Thus, we ask the reviewer to relate to the absence of other analyses techniques with an understanding. Your helpful suggestions will become the aim of our future articles.
Also, we corrected our paper according to your minor comments. Below you can see our answers:
- We improved the equations for a better understanding of the reader.
- You were absolutely right about our mistake. We corrected it.
- We added a table which shows all the values of shear strength and crystallinity for samples manufactured at a pressure of 25 or 50 MPa and a temperature range from 155 °C to 170 °C.

Reviewer 2 Report
In this work, the authors report the fabrication and characterization of a composite material based on ultra-high molecular weight polyethylene.
The article is clear and the quality of the presentation very good. I have only one suggestion:
1) Modify the article title in a way that doesn't need an acronym.
Author Response
Response to Reviewer 2 Comments
In this work, the authors report the fabrication and characterization of a composite material based on ultra-high molecular weight polyethylene.
The article is clear and the quality of the presentation very good. I have only one suggestion:
1) Modify the article title in a way that doesn't need an acronym.
Response 1: Please provide your response for Point 1. (in red)
Dear reviewer, thank you for carefully reading the paper and giving good feedback! We changed the title according to your suggestion, so the title has no acronyms anymore.

Reviewer 3 Report
In the paper 'The hybrid self-reinforced composite materials based on UHMWPE', the authors presented a study on the correlation between hot compaction pressure /temperature and mechanics of polymer composites. The reviewer have the following comments
- In the introduction, please elaborate on 'self-reinforced composite'. The concept was not well-defined in the paper.
- Figure 6, the authors should present all SEM images with identical scale bar, if possible.
- please report the results in the format of average +- standard deviation, which is missing in the text.
- What is the thickness of the remelted fiber out layer? would this layer (illustrated in figure 2b) be visualized in SEM?
- In conclusion part, the authors suggested that 'the adhesion between layers increase with an increase in matrix content'. The structure change also affect flexural strength and shear strength. How does the special etching approach influence fracture resistance of the hybrid material? The surface morphology shown in Figure 6, could the surface texture result from the fracture (ductile versus brittle) instead of the treatment method?
- The author mentioned that 'the hot compaction allows to melt only surface of each fibers'. Any evidence to support the statement? It wasn't clear whether the cross-sectional or the peripheral of the individual fibers were melted.
Author Response
Response to Reviewer 3 Comments
In the paper 'The hybrid self-reinforced composite materials based on UHMWPE', the authors presented a study on the correlation between hot compaction pressure /temperature and mechanics of polymer composites. The reviewer have the following comments
- In the introduction, please elaborate on 'self-reinforced composite'. The concept was not well-defined in the paper.
- Figure 6, the authors should present all SEM images with identical scale bar, if possible.
- please report the results in the format of average +- standard deviation, which is missing in the text.
- What is the thickness of the remelted fiber out layer? would this layer (illustrated in figure 2b) be visualized in SEM?
- In conclusion part, the authors suggested that 'the adhesion between layers increase with an increase in matrix content'. The structure change also affect flexural strength and shear strength. How does the special etching approach influence fracture resistance of the hybrid material? The surface morphology shown in Figure 6, could the surface texture result from the fracture (ductile versus brittle) instead of the treatment method?
- The author mentioned that 'the hot compaction allows to melt only surface of each fibers'. Any evidence to support the statement? It wasn't clear whether the cross-sectional or the peripheral of the individual fibers were melted.
Response 1: Please provide your response for Point 1. (in red)
Dear reviewer, thank you for carefully reading of the paper and your helpful suggestions. We corrected the paper according to your comments; see below for a more detailed description. English was improved through the paper.
- The idea of self-reinforced composite material was described in detail on the lines from 62 to.73
- Unfortunately, preparing these samples via cutting and etching is complicated. We tried to show the most appropriate pictures of the structural features.
- The deviations were calculated for all the measured values added in the paper. Also, all graphs were corrected according to recalculated data.
- The the thickness of the remelted fiber was shown in figure 6. The samples were cut perpendicular to fibers’ axis and the resulted cross sections were etched with potassium permanganate solution that proved more effective etching for remelted part of each fiber. In figure 6 etched grooves is directly related to the remelted fibers layer (matrix of the composites). As it can be seen, the thickness of the matrix layer is in the range of 1-3 µm depending on hot compaction parameters.
- We assume that you misunderstood us. Only the hot compaction parameters influence on the composites’ properties because the hot compaction affected matrix-to reinforced ratio. We do not expose the initial fibers to any treatment. The etching was only used to SEM visualization to study the effect of hot compaction to structure of the composites. The surface morphology shown in Figure 6 is an etched surface, not a fracture surface. Unfortunately, we do not study a fracture resistance of the composites.
the matrix on the surface of sample which was cut in liquid nitrogen.
- It is well known that the potassium permanganate etching is more effective for isotropic UHMWPE and, as was mentioned earlier, in figure 6 etched grooves is directly related to the remelted fibers layer. Thus, we can conclude that hot compaction allows to melt only surface layer of each individual fiber. Below, you can see the etching mechanism in the self-reinforced composites.
Figure 1 – scheme of SRC’s part in plane parallel to fibers’ axis (a) before and (b) after the etching
Also, in your previous paper [Zherebtsov, D.; Chukov, D.; Torokhov, V.; Statnik, E. Manufacturing of Single-Polymer Composite Materials Based on Ultra-High Molecular Weight Polyethylene Fibers by Hot Compaction. Journal of Materials Engineering and Performance 2020.] we showed the SEM images that prove that 'the hot compaction allows to melt only surface of each fibers'.

Round 2
Reviewer 1 Report
I recommend to publish the article for publication in present form.
Author Response
Thank you for your efforts! Your advice helps us to make our article better!
Reviewer 3 Report
Thank you for revising the manuscript. The reviewer do not have additional comments.
Author Response
Dear reviewer, thank you for carefully reading of the paper and your helpful suggestions!